

# Captive breeding in the endangered alpine tree frog, *Litoria verreauxii alpina*

Laura A. Brannelly[1], Preeti Sharma[1,2] and Danielle K. Wallace[1]

[1] Melbourne Veterinary School, University of Melbourne, Werribee, Victoria, Australia
[2] School of Environment and Science, Griffith University, Southport, Queensland, Australia

## ABSTRACT

Amphibians are experiencing dramatic worldwide declines and many species are reliant on captive breeding programs to ensure continued survival. However, captive breeding in amphibians is not always successful because many species, especially ones in decline, have particular and specific breeding needs. The endangered alpine tree frog, *Litoria verreauxii alpina*, has never been bred in captivity before. Due to its dramatic declines across the Australian Alps caused by the global pandemic chytridiomycosis, the species is a potential candidate for captive assurance colonies, which rely on captive breeding. For this study we tested hormone induction using two hormones that have had some success in other amphibian species, to no avail. We then tried outdoor breeding mesocosms during the winter/spring at temperatures similar to their natural breeding season, which was successful. Sixty-five percent of the egg masses laid successfully hatched tadpoles. Females laid more than one clutch over the experiment indicating either a shorter than annual ovulation cycle, or that females are capable of partial ovulation during breeding events. Outdoor breeding mesocosms are a possibility outside the native climate of a species, provided that temperatures overlap with their natural environment. Here, we highlight that troubleshooting is essential before embarking on a captive breeding program of a species that has not been bred before. Hormonal induction of breeding is not always successful; therefore, outdoor mesocosms might be required to achieve healthy tadpoles.

## INTRODUCTION

Amphibians are experiencing some of the most devastating population declines on the planet (*Houlahan et al., 2000*; *Stuart et al., 2004*; *Collins, 2010*). Many species around the world require intensive conservation measures to protect against extinction. However, captive breeding of amphibians can be a challenging endeavor because species often rely on specific and particular environmental stimuli to induce breeding, which can be difficult to simulate under captive conditions (*Kouba, Vance & Willis, 2009*; *Silla & Byrne, 2019*). Assisted reproduction technologies like exogenous hormones can be effective at reducing the need for particular abiotic or biotic stimuli and induce breeding behaviours in captivity (*Silla & Byrne, 2019*). There are two commonly used exogenous hormones that induce breeding in amphibians, gonadotropin-releasing hormone agonist (GnRH-a)

Corresponding author
Laura A. Brannelly,
laura.brannelly@unimelb.edu.au

and human chorionic gonadotropin (hCG) (*Silla & Roberts, 2012*; *Brannelly et al., 2015*; *Silla, McFadden & Byrne, 2019*). Both of these exogenous hormones are used to elicit spermiation in males and spawning in females (*Mansour, Lahnsteiner & Patzner, 2009*; *Vu, Weiler & Trudeau, 2017*; *Silla & Byrne, 2019*). In some species, hormone induction will result in spontaneous egg laying in females, such that sperm and eggs can be collected separately and fertilized in the laboratory (*Mansour, Lahnsteiner & Patzner, 2009*; *Wlizla et al., 2017*). In other species, spawning is enhanced by hormone induction, but normal mating behaviours between males and females are still required (*e.g.*, amplexus) (*Trudeau et al., 2013*; *Vu, Weiler & Trudeau, 2017*; *Brannelly, Ohmer & Richards-Zawacki, 2019*).

Finding appropriate captive breeding protocols for amphibian species can be challenging because species often respond differently to different methods. Protocols have been refined and there are reliable methods for collecting sperm and eggs in model species like *Xenopus laevis* and *X. tropicalis* (*Mansour, Lahnsteiner & Patzner, 2009*; *Wlizla et al., 2017*). Whereas, protocols for non-model species are less reliable and require species-specific troubleshooting (*Silla & Roberts, 2012*; *Clulow et al., 2018*; *Pham & Brannelly, 2022*). In other amphibian species, researchers have been unable to identify exogenous hormones that are successful at inducing gamete release and spawning, or there are protocols for males but not for females (*Clulow et al., 2018*). In cases where exogenous hormones are ineffective, amphibian captive breeding colonies might require specific environmental stimuli to induce breeding. In these cases, outdoor enclosures might offer a sustainable alternative where animals can court and spawn under semi-natural conditions, and then fertilized egg masses can be retrieved.

The alpine tree frog, *Litoria verreauxii alpina*, is endemic to the alpine regions of New South Wales and Victoria, Australia and is considered endangered. The species has declined dramatically since the 1980s, and has been extirpated from greater than 80% of its former distribution (*Gillespie et al., 2015*). The primary cause of decline in this species is the deadly disease chytridiomycosis, caused by the fungal pathogen, *Batrachochytrium dendrobatidis* (*Brannelly, Scheele & Grogan, 2020*). The eight to 10 populations that remain are physically isolated from each other with virtually no possibility of geneflow among the populations (*Banks et al., 2020*). However, these remaining populations are seemingly stable, where adults breed and recruitment is successful each year. The dramatic declines throughout its range and restricted current distribution makes this species an ideal candidate to test captive breeding protocols. There are no current captive assurance colonies of *L. v. alpina*, yet it is well understood that there is a risk of further declines due to climate change and other factors (*Brannelly et al., 2016b*; *Scheele et al., 2016*). Wildfire frequency and intensity as well as drought are a high risk in the Australian Alps (*Whetton, Haylock & Galloway, 1996*) and could decimate the remaining populations. Identifying reliable methods for captive breeding are critical to the longevity of captive assurance colonies as well as research captive colonies (*Clulow, Trudeau & Kouba, 2014*; *Clulow et al., 2019*). Having a healthy and sustainable captive colony allows for improved management capacity and high quality research on endangered species without threatening current population stability.

The aim of this study was to test methods for captive breeding in the alpine tree frog. Breeding under laboratory conditions can be challenging in *Litoria* species, therefore we
trialed two different methods: indoor laboratory controlled conditions using exogenous hormones following brumation, and an outdoor breeding mesocosm following brumation. In the first year we tested if a captive environment alone could produce breeding behaviours (no-hormone control), and also tested the efficacy of two exogenous hormone types (hCG [Chorulon]; GnRH-a [Lucrin]) on breeding behaviours and reproductive success under stable laboratory conditions. In the second year, we tested the efficacy of using outdoor enclosures without exogenous hormone assistance on reproductive behaviours and reproductive success. Identifying an effective and reliable method for captive breeding in *L. v. alpina* is critical for the sustainability of this species under captive conditions. Furthermore, understanding captive breeding in this species can help direct captive breeding in other similar species.

## METHODS

### Animal husbandry

*Litoria verreauxii alpina* were collected as tadpoles ($n = 200$) from the wild in December 2018 from one site. The collection site had no canopy cover and little submerged vegetation. The site location is site redacted because of their endangered status, but it was a man-made pond that is drained annually (end of spring/early summer). The tadpoles were transported to the University of Melbourne and raised until adulthood. They were housed communally in enclosures with gravel and moss substrate and watered daily with carbon filtered tap water. Frogs were fed gut loaded and vitamin dusted crickets *ad libitum* 3–4 times weekly at 16–18 °C and maintained at a 12 h:12 h light:dark cycle. All animals in the captive colony at the time of the study were utilized in this study.

Litoria v. alpina lives in the Australian Alps and breeds for 6–12 weeks (*Brannelly et al., 2015*; *Brannelly et al., 2016a*; *Brannelly, Scheele & Grogan, 2020*) during the early spring snow melt, following an overwintering period where they brumate in the leaf litter in forest habitat near the breeding ponds. Overwintering in this species is thought to be a physiological requirement to successfully breed, particularly for gravid females to release their eggs. Any offspring produced in this study entered the captive colony. No animals were euthanized, and investigators were not blinded to the treatments provided.

### Overwintering

Eleven months after being brought into captivity, and 9–10 months after completing metamorphosis, animals were placed in smaller communal 1L enclosures (3–4 animals per enclosures) with a damp moss substrate and artificially overwintered for 12 weeks. Frogs were haphazardly chosen and separated by sex ($n = 31$; 17 females and 13 males). Animals were weighed to the nearest 0.01 g, and their snout to vent length (SVL) was measured to the nearest 0.1 mm using dial calipers. Temperature was reduced at a rate of 1 °C per day until it reached 4 °C. Animals were maintained at 4 °C for 10 weeks. Enclosures were misted with filtered water daily/as needed, and the substrate was replaced every 4 weeks. Animals were not fed during this brumation period (as is usual with brumating amphibians) and were maintained in the dark. After 10 weeks of brumation, animals were slowly warmed to 18 °C at a rate of 1 °C per day. Morphometric data including mass and length was

taken again following brumation. Male and female *L. v. alpina* that develop in captivity are often sexually mature at 9 months after metamorphosis (*Brannelly et al., 2021*). All females appeared to be gravid after brumation. Males had visible and pronounced nuptial pads and dark throat colouration, indicating that all animals were sexually mature.

### Indoor breeding enclosures

Immediately following overwintering, animals were placed in 50 L breeding mesocosms, containing aquatic as well as terrestrial habitat. The mesocosm was made of clear plastic and contained approximately 30 L of water with plastic foam floats affixed to one side of the enclosure to provide terrestrial habitat. Artificial plants were placed in the aquatic habitat to mimic aquatic vegetation where the frogs would lay their eggs in the wild. A total of 2–4 L of water was removed and replaced each day. Animals were fed gut loaded and vitamin dusted crickets *ad libitum* once weekly while in their breeding mesocosms. Breeding mesocosms contained 3–5 frogs per mesocosm and were housed in laboratory conditions maintained at 16–18 °C, and a window in the laboratory to provide some natural light during the day.

### Induction with exogenous hormones

Two different hormone treatments were tested ($n = 31$ animals in total, individually haphazardly chosen per group): GnRH-a (three mesocosms; 2F:2M, 2F:2M; 2F:1M) and hCG (three mesocosms; 2F:2M, 2F:2M, 2F:1M), along with no-hormone controls (two mesocosms, 3F:2M, 3F:1M). We gave the females a priming dose 24 hrs after entering the breeding enclosures, and 48 hrs before the induction dose while the males were given only an induction dose. Males and females were given an induction dose at the same time. Doses given were set for each individual, regardless of mass. The priming dose for GnRH-a (leuprolide acetate; Lucrin® Depot, AbbVie Limited) was 0.5 µg and the induction dose was 8 µg. Both doses were diluted in 200 µL of bacteria-filtered amphibian ringers solution (*Llewelyn, Berger & Glass, 2019*). The hCG priming dose was 12 IU (Chorulon®, Intervet Inc.) and the induction dose was 54 IU Chorulon diluted in 200 µL of bacteria-filtered amphibian ringers' solution; the no hormone control group received a priming and ovulatory injection of 200 µL bacteria-filtered amphibian ringers' solution (no hormones added to the vehicle). Animals were injected ventrally into their intracelomic cavity with a 27 gauge needle.

After the first induction dose, we attempted to strip the females of their eggs for a maximum of 10 min, 4 times a day over 3 days. To encourage spawning, we gently massaged the female frogs in the abdomen, and inserted a 10 uL pipette tip into the cloaca. We continuously played a 15 min chorus of *L. v. alpina* on loop in the background to encourage breeding during the trial. Males and females were monitored in their breeding mesocosms for a total of 30 days. Instances of calling and amplexus were recorded in each tank at least once per day. If no eggs were laid one week following the induction dose, all animals from all three treatment groups (GnRH-a, hCG and the no-hormone control) were given another induction dose and monitored. A total of three induction doses were given if spawning did not occur. After 30 days in total, animals were removed from their breeding mesocosms and placed back in their normal communal enclosures.
## Outdoor breeding mesocosm trial

Twelve months after the induction trial concluded, a subset of animals ($n = 24$) entered brumation again following the same protocol as described above. After 8 weeks of brumation at 4 °C, 12 individuals (six female and six male) were brought out of brumation with a slow increase in temperature of 1 °C per day until 11 °C. Mass and SVL was recorded before and after brumation, and then immediately following brumation the animals were placed in outdoor breeding mesocosms. Hormone induction was not used in this outdoor breeding trial because it was unsuccessful in the first trial. The 50 L clear plastic containers were placed in an open-air aviary that allowed natural sunlight and weather to enter the mesocosms. Mesocosms consisted of both terrestrial and aquatic habitat, with approximately 20 L of water and gravel terrestrial substrate piled to one side of the enclosure. Artificial plants were affixed to the terrestrial side of the enclosure to provide hides, and artificial aquatic plants and zip ties were placed in the aquatic side of the enclose to mimic aquatic vegetation where the frogs would lay their eggs in the wild. Animals were placed in the outdoor mesocosms when the air temperature was 11 °C in winter (June in Melbourne, Victoria) and maintained outside throughout the winter and spring. The rest of the animals ($n = 12$; nine females and three males) remained in brumation for 13 weeks in total. These animals were placed in the outdoor breeding mesocosms when the air temperature was 9 °C (July).

Animals were maintained in three breeding mesocosms. When the first 12 animals were brought into the outdoor enclosures, two females and two males were placed in each enclosure. When the remaining animals entered the breeding mesocosm, the total number of animals per mesocosm was eight, with each mesocosms containing five females and three males per enclosure. Animals were played a recorded 15 min chorus of *L. v. alpina* playing on loop in the background at all times. Each enclosure was checked daily for instances of calling, amplexus, and egg masses. Animals were fed with crickets *ad libitum* once per week and tanks were flushed with 10 L of aged tap water daily. At three points throughout the study (days 68, 100 and 130 after the first animals entered the outdoor mesocosm) animals were redistributed across the tanks to help promote spawning. On day 100 of the experiment, nine females who were no longer visibly gravid were moved to a single mesocosm without a male, while the remaining six females and nine males were split across two mesocosms (ratio of 3F:4M, 3F:5M). Adult frogs were maintained in the breeding mesocosm, and breeding activities were monitored through the spring (November; 19–24 weeks). On day 169 of the experiment, animals were removed from the outdoor mesocosms, measured for SVL and mass, and returned to the laboratory colony.

When egg masses were laid, they were removed and brought into the laboratory. Egg masses were placed in 5 L containers with aerated carbon filtered water. Half of the water was replaced once per week. When masses hatched and tadpoles became free swimming, they were removed and placed in 50 L aquatic enclosures. Hatched tadpoles and unhatched eggs were counted for egg mass size and the proportion of the eggs that were fertilized/viable. Egg masses were monitored for up to 14 days and if they did not develop, they were discarded.

## Ethics and permits

The work was performed under the University of Melbourne animal ethics application 10267, State of Victoria Department of Environment, Land, Water and Planning Wildlife Act 1975 wildlife research permits 180705 and 10010126.

## Statistical analyses

All analyses were conducted in R in the RStudio interface (*RStudio Team, 2016*; *R Core Team, 2017*). To explore the impact of hormone treatment on the probability of mating display (amplexus pairs or calling males) within a hormone treatment group we used a generalized linear model (GLM) with a binomial distribution ($n = 34$ observed displays), where mating display (whether amplexus or calling was recorded within a treatment during the daily checks, or not) was the response variable, the predictor variables were hormone treatment (no-hormone control, GnRH-a-treated and hCG-treated) and days since last injection (injections occurred on day 0, 8, and 13).

To explore the effect of time since the animals were placed in outdoor breeding enclosures on whole egg mass viability, we conducted a GLM with a binomial distribution where egg mass viability (0 tadpoles hatched: at least one tadpole hatched, $n = 23$ egg masses laid) was the response variable, and day of egg mass lay was the predictor variable. To assess the effect of days in the outdoor breeding enclosure on the size of the egg mass laid we conducted a multinomial logistic regression (MR) using 'multinom' within the package 'nnet' (*Ripley & Venables, 2022*), where the size of the egg mass laid (small = <100 eggs; medium = 100–300 eggs; large = >300 eggs) was the response variable and days in outdoor mesocosm was the predictor variable. To explore the correlation of egg mass size and days in the outdoor mesocosm had on the proportion of eggs within the egg masses that hatched, we used a linear model (LM) where percent of eggs that hatched was the response variable, and the predictor variables were days in outdoor mesocosm and size of the egg mass laid. Because we could not match individual female with the egg masses laid, we did not include female size as a fixed effect in this analysis.

To understand how brumation and breeding influenced the mass of the animals we conducted LMs, where mass of the animal in grams was the response variable and timepoint in the experiment (before brumation, at the end of brumation/the start the outdoor breeding experiment, and at the end of the breeding experiment) was the response variable. Sex of the animal is a confounding effect: females are larger than males when they are sexually mature; therefore, male ($n = 26$ datapoints analysed) and female ($n = 46$ datapoints analysed) mass was analysed separately. Animals were not marked or tagged during the experiment so mass change is based on the timepoint, not the individual. We conducted Tukey's post hoc tests using the package 'emmeans' (*Lenth et al., 2022*), and assessed effect size using Cohen's $d$ statistic where appropriate. Model outputs were assessed for model assumption violations to ensure appropriate analyses were conducted. No animals or mesocosms were excluded from analysis.

## RESULTS

### Induction with exogeneous hormones

While all females were visibly gravid following brumation, no eggs were released, either through the attempted stripping procedures or through amplexus with male frogs. Although no spawning took place, we found signs of breeding activity in all hormone-treated groups. No-hormone control animals did not display signs of breeding activity. GnRH-a-treated animals displayed significantly more reproductive displays (amplexus, calling males) than the hCG-treated animals (GLM: $\chi^2 = 11.550$, $p = 0.003$). Over the four-week breeding trial we recorded 13 instances of amplexus and 3 instances of calling in GnRH-a-treated animals. Breeding activity appeared to be concentrated on or one day after the hormone was injected (Fig. 1); however, days after injection was not a significant effect on the probability of a mating display (GLM: $\chi^2 = 1.114$, $p = 0.291$). While GnRH-a-treated animals displayed breeding activity, no spawning events occurred, even after extended amplexus. We recorded one instance of amplexus (one day after the first injection) and one instance of male calling (one day after the second injection) in the hCG-treated animals. Breeding activity did not occur in any treatment after the third injection.

### Outdoor breeding

In the 169 days that animals inhabited their outdoor breeding mesocosm enclosures, 23 distinct egg masses were laid by the 15 females placed in the enclosures. Of those 23 laid masses, 15 masses were viable (65%; at least one tadpole successfully hatched from the mass). The first egg mass was laid on day 23, and the last viable egg mass was laid on day 163 (Fig. 2). No egg masses contained 100% viable eggs, but 20% of the viable egg masses (three masses) hatched over 75% of eggs laid. 13.33% of the viable egg masses (two masses) had less than 25% of eggs hatch. The majority of viable egg masses laid had between 25% and 75% hatching success and were laid in the middle of the experimental time frame (day 65–90; Fig. 2).

On day 100, nine females that were no longer visually gravid were moved to a female only mesocosm. After that shift in sex ratio (from 6F:3M in each tank to 3F:5M and 3F:4M per tank) nine egg masses were laid (Fig. 2). Four of those egg masses came from the tank with only females.

There was no effect of day since animals were placed in outdoor breeding enclosures on the viability of the egg masses (GLM: $\chi^2 = 0.348$, $p = 0.556$), or the size of the egg mass laid (MR: $\chi^2 = 04.463$, $p = 0.107$). There was no effect of day or size of the egg mass laid on the proportion of eggs that were viable within the egg mass laid (LM: day, $F_{1,19} = 0.113$, $p = 0.740$; egg mass size, $F_{2,19} = 1.767$, $p = 0.198$).

Neither males nor females lost a significant amount of body mass during their brumation period, but both males and females lost significant body mass between brumation at the end of the experiment (Fig. 3). Female frogs lost 43.4% of their body mass (6.9 ± 1.2 g before brumation, 3.9 ± 0.8 g after breeding; $d = 2.88$) from the time they entered brumation to the end of the experiment, and males lost 26.7% of their body mass (4.5 ± 0.9 g before brumation, 3.3 ± 0.7 g after breeding; $d = 1.49$) (LM: Females; $F_{2,43} = 30.786$, p <0.001: Males; $F_{2,23} = 4.929$, $p = 0.017$).

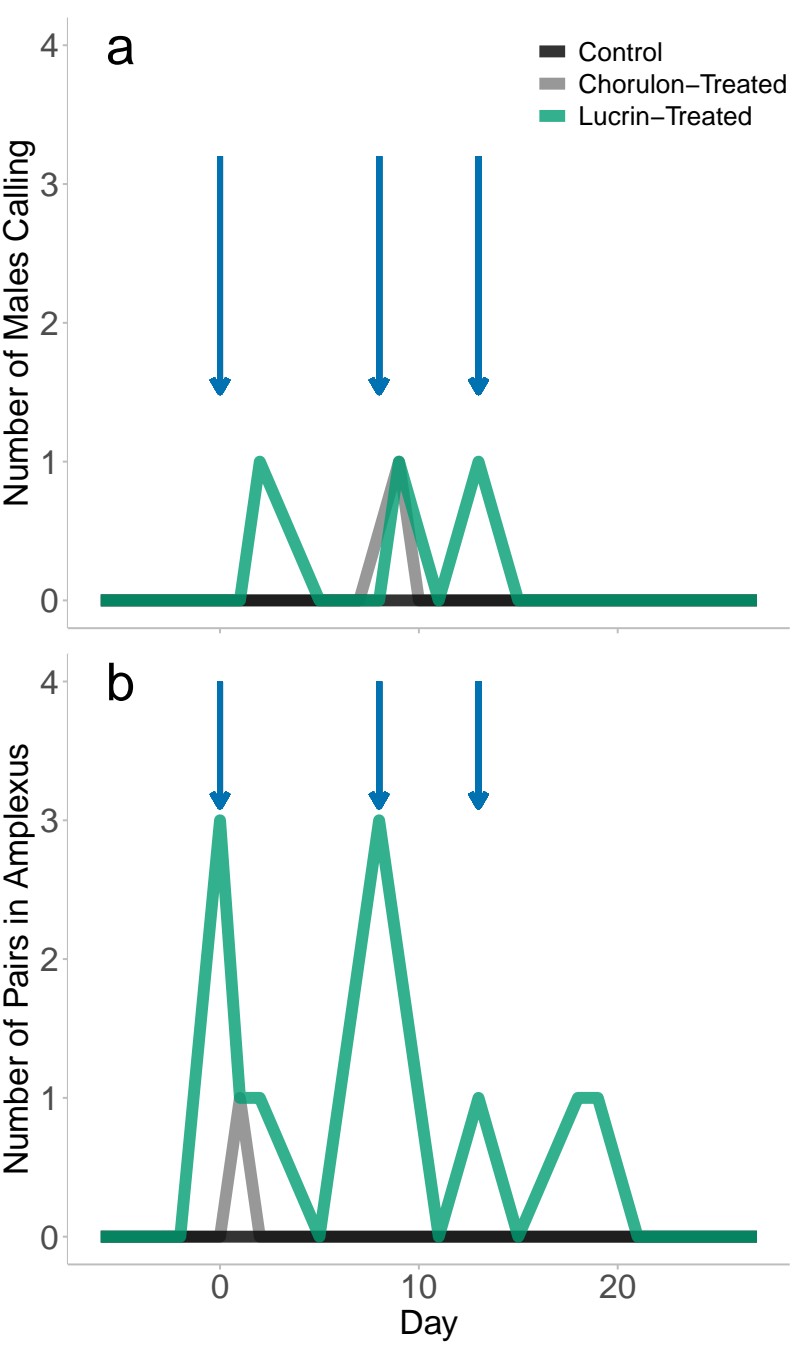

**Figure 1  The breeding behaviour observed following exogenous hormone injections.** The blue arrows indicate when hormone injections took place (day 0, 8, and 13). There was a total of five males per exogenous hormone groups, and 3 males in the no-hormone control group. Animals were checked once daily, and number of calling males and pairs in amplexus that were observed per treatment group during the daily checks were recorded.

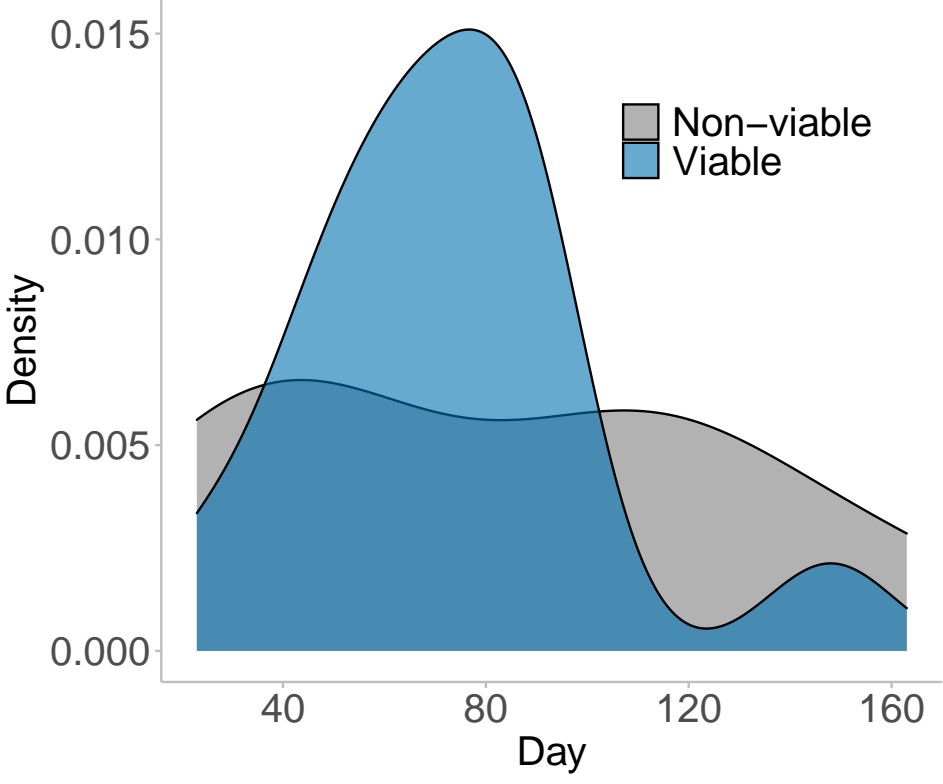

**Figure 2** **The viable and non-viable egg masses laid over the experimental timeframe.** A density plot depicting the viable and non-viable egg masses laid over the experimental timeframe. A total of 23 distinct egg masses were laid, 15 of which were viable—where at least one tadpole hatched from the clutch. These 23 egg masses were laid between day 23 and day 163 of the experiment. 12 animals entered the breeding mesocosm on day 0 (six males and six females), and the remaining 12 animals entered the breeding meso-cosm on day 40 (three males and nine females). All animals were collected, and the experiment ended on day 169. Note there was no significant difference in when viable and non-viable egg masses were laid (GLM: $\chi^2 = 0.348$, $p = 0.556$). The male: female ratios within the mesocosms shifted on day 100 from 5F:3M for all three mesocosms to 9F:0M, 3F:5M, and 3F:4M.

## DISCUSSION

The hormone induction methods trialed here using Lucrin (GnRH-a) and Chorulon (hCG), as well as the no-hormone control, were unsuccessful at inducing spawning. *Litoria* species are notoriously difficult to induce spawning in a captive indoor setting (*Clulow et al., 2018*); therefore, this result was not surprising. Since this trial we have begun optimizing an effective hormonal induction dose for male *L. v. alpina* in captivity, where an effective dose to induce spermiation using GnRH-a was lower (1.5 µg GnRH-a per animal) than trialed here (8 µg per animal), and hCG dose was higher (120 IU per animal) than trialed here (54 IU per animal; *Pham & Brannelly, 2022*). The dose of GnRH-a used in this study did not produce sperm in a later study (*Pham & Brannelly, 2022*), yet it did result in male breeding behaviour like amplexus and calling. These behaviours without sperm release might indicate that the physiological mechanism for breeding behaviour

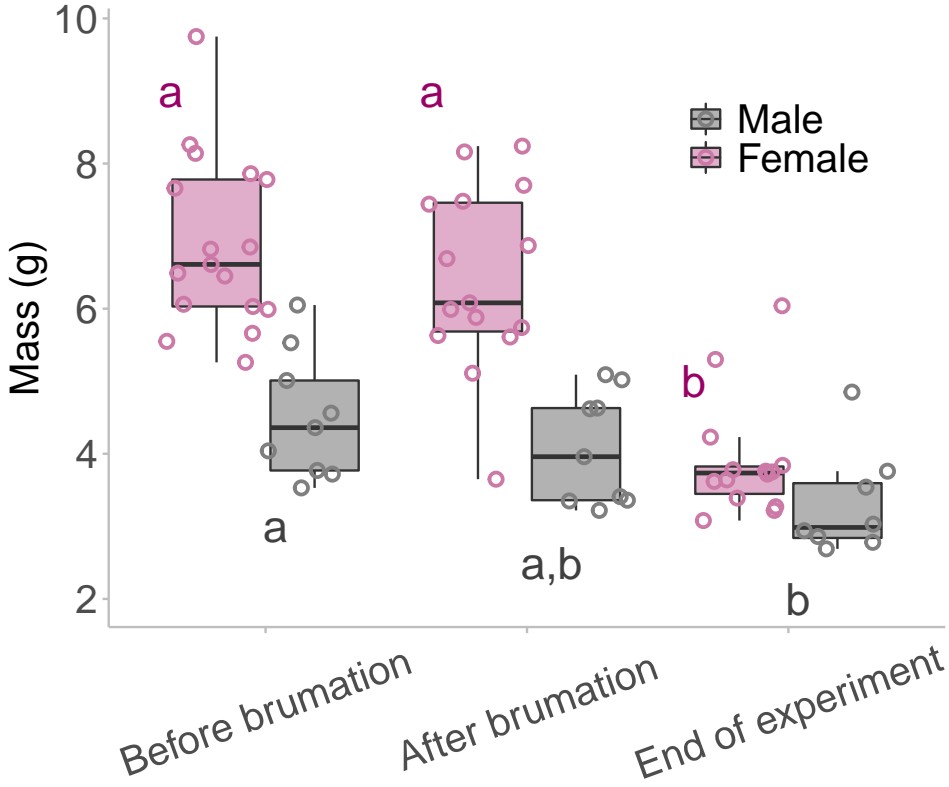

**Figure 3  Mass of the animals taken across the outdoor breeding mesocosm experiment.** Mass of the animals taken before they began overwintering/brumation, when they were removed from the overwintering/brumation incubator and placed in the breeding mesocosm, and at the end of the experiment. The middle lines of the box plots represent the median mass for that timepoint, the boxes represent the 1st and the 3rd quartile, and the whiskers represent the 95% confidence interval. The points represent the individual masses of the animals per time point. Male and female mass was analysed separately; therefore, the "a" and "b" annotations represent statistical differences between the mass of the animal at different times of the experiment, within a single sex: pink for female and grey for the males. Statistical significance between timepoints was determined using Tukey's *post hoc* tests.

and sperm production is different; therefore, an optimum dose of hormonal induction would need to induce both breeding behaviour and spermiation. There is more work to be done improving the use and dose of exogenous hormones in *L. v. alpina*, particularly for inducing ovulation in females (*Clulow et al., 2018*; *Pham & Brannelly, 2022*).

We successfully bred *L. v. alpina* in outdoor mesocosms following an artificial brumation period at 4 °C for 8–13 weeks. The temperature in Melbourne, Victoria, Australia during winter is approximate to the temperature in the Australian Alps during spring when breeding would naturally occur (average of 5–17 °C over a 24 hr period; *Brannelly et al., 2015*); therefore, it is possible to captively breed these animals following the protocol used in Victoria, Australia. However, because egg masses were laid over a period of four months, adapting this protocol to a warmer climate might be less successful because this species is cool-adapted and would require a shorter breeding window. Because we were able to breed animals in outdoor enclosures but not in the lab with or without hormone

induction there is likely one or several environmental parameters (such as temperature, humidity, rain events, barometric pressure) that encourage the animals to breed. In this study we did not collect detailed environmental data during the breeding experiment, but this would be important to know if which weather events are important to encourage breeding within their natural environment and outside their natural environment (*e.g.*, our facility in Melbourne), and this could then be adapted for indoor breeding facilities. Although, sometimes environmental parameters are not enough to encourage breeding: in other species, like *Rana pipiens*, brumation and then placing the animals in outdoor breeding mesocosms within their natural habitat range was not enough to induce breeding: hormonal induction was required (*Trudeau et al., 2013*; *Vu, Weiler & Trudeau, 2017*). Captive breeding requirements are species specific, and troubleshooting protocols are necessary for captive breeding to be successful.

Because the animals were housed in communal mesocosms it is unclear which females laid egg masses, which did not lay, and which ones laid multiple egg masses. Egg masses were laid over 140 days; it is possible that females who spawned early in the experiment developed new eggs by the end of the experiment. Female frogs with seasonal breeding cycles often regenerate eggs once per year, but some frog species do breed at multiple times throughout the year (*Rastogi et al., 2005*). The sympatric species, *Crinia signifera*, is known to occasionally lay multiple egg clutches within a single breeding season when ecological conditions will allow (*Lemckert & Shine, 1993*), indicating that multiple clutches within a single female is possible in species that share the habitat.

In amphibian species with long or continuous breeding cycles the ovaries often have eggs at multiple stages of development at any one time, while in species that have an annual duration ovarian cycle, the eggs within the ovary mature at approximately the same time (*Rastogi et al., 2005*). For *L. v. alpina*, the previous work on oogenesis indicates asynchronous follicular development (*Brannelly et al., 2016b*; *Brannelly et al., 2021*), such that it might be physiologically possible for one animal to lay multiple egg clutches within the duration of this experiment. Some species are known to lay two egg clutches within a short timeframe (30 days) (*Rastogi et al., 1983*). A previous experiment on *L. v. alpina* noticed increased oogenesis after exposure to the pathogen *Batrachochytrium dendrobatids* after just 3–12 weeks (*Brannelly et al., 2016b*), suggesting that oogenesis can occur more quickly than on an annual cycle.

The short breeding season and annual ovulatory cycle in *L. v. alpina* is likely due in part to the limiting environmental conditions of the Australian Alps and infectious disease causing high adult mortality before the breeding season would historically end (*Brannelly et al., 2015*; *Brannelly, Scheele & Grogan, 2020*; *Scheele et al., 2015*). *Litoria. v. alpina* is part of species complex with species *L. ewingii, L. paraewingii,* and *L. v. verreauxii*, which all have an extended breeding season and can breed throughout much of the year (*Hero, Littlejohn & Marantelli, 1991*; *Lauck, Swain & Barmuta, 2005*; *Smith et al., 2013*; *Scheele et al., 2014*). Retaining an asynchronous breeding capacity might be due to their phylogenetic history rather than their seasonal breeding ecology.

It is also possible that the distinct egg masses represent partial ovulations for the females. Partial ovulation of mature eggs seems to be relatively rare in natural amphibian

populations, but some captive breeders have experienced partial ovulation in female frogs particularly following hormone inductions (*Lehman, 1977*; *Jorgensen, 1984*; *Sive, Grainger & Harland, 2000*). Egg clutches resulting from partial ovulation can result in poor fertilization success, reduced egg viability, or few tadpoles hatching (*Silla & Byrne, 2021*). For example, the species *Heleioporus eyrei* only partially ovulated following hormonal inductions and had much lower early embryo survival compared to other species that ovulated completely following hormonal induction (*Silla & Byrne, 2021*). The egg masses collected in this study where overall much smaller than expected for this species (a range of 50–400 eggs per clutch with many clutches being approximately 100–200 eggs per clutch), which typically lays 300–400 eggs per clutch in the wild (unpublished data; *Gillespie, Osborne & McElhinney, 1995*). Furthermore, egg mass viability was low, with 43% of the masses producing no or less than 25% viable tadpoles. However, the low hatching success rate of the majority of the egg masses produced in this experiment is consistent with the low viability of other studies that explored methods of captive breeding (*McFadden et al., 2008*; *Trudeau et al., 2013*; *Vu, Weiler & Trudeau, 2017*; *Brannelly, Ohmer & Richards-Zawacki, 2019*). It is understood that eggs and egg masses in captive breeding colonies are less viable than they would be in the wild, although this is not universally the case (see *Reichling et al., 2022*).

When we moved the females that were no longer visibly gravid into a female-only mesocosm, four unviable egg masses were laid in that tank. We are confident that those four non-viable clutches came from females that already laid, but we are unsure whether these spawning events represent new ovulation or a partial ovulation. Gravid females were determined visually, if eggs could be seen though a clear patch in their groin skin, but this is not a fool-proof method. Female frogs injected with exogenous hormones will spontaneously spawn (*Mansour, Lahnsteiner & Patzner, 2009*; *Wlizla et al., 2017*), but it is unclear why females would spontaneously spawn under the conditions in this trial. Male mating calls can elicit a spawning position in some female frog species (*Knorr, 1976*), such that calling and other breeding stimuli might elicit spontaneous spawning. Spontaneous spawning without amplexus can occur in frogs (*Jungfer & Weygoldt, 1990*), and we have seen it in our captive facility in *L. v. alpina* and *L. ewingii* (LA Brannelly, pers. obs. 2022). Spontaneous spawning might be a mechanism for the animals to reduce their risk of becoming egg-bound when there are no prospective breeding opportunities and they cannot reabsorb the eggs. Or it could occure following stimulation from calling males. If *L. v. alpina* are spontaneously spawning due to breeding stimuli, it might indicate that they are a good species for exogenous hormone trials for female spawning. It is clear that the hormones and doses used in this study were unsuccessful, but more research could be done to try to identify a more successful dose—possibly using hCG at a higher dose because it was successful in producing spermic urine in males (*Pham & Brannelly, 2022*).

Notably, the animals used in the outdoor breeding experiment were the same animals that underwent the failed hormone induction trial the year before. All females remained visibly gravid over that year of recovery from the hormonal induction trial, but it is possible that the mature eggs present in the ovaries atrophied by the time the outdoor mesocosm experiment began. Poor quality of eggs due to atrophy or follicular atresia

(*Rastogi et al., 2005*) might explain the low viability observed in many of the egg masses laid. However, captive breeding frequently produces egg masses that are less than 100% viable (*Trudeau et al., 2013*; *Vu, Weiler & Trudeau, 2017*; *Brannelly, Ohmer & Richards-Zawacki, 2019*); therefore, the low viability might be due to captive breeding methodology and/or suboptimal eggs. To determine if the egg quality was poor due to the prior hormonal treatment, a follow up study should follow the same methods but use animals that had not been used in a breeding experiment before.

Both females and males lost a substantial amount of body mass during the outdoor breeding experiment. There is no data from wild *L. v. alpina* comparing pre- and post-breeding mass to assess if this loss in mass is typical. Female frogs of a variety of species tend to lose 20–30% of their body mass following spawning (*Ryser, 1989*; *Lemckert & Shine, 1993*; *Rastogi et al., 2005*), yet we observed a loss in mass of over 40% between brumation and the end of the experiment. Animals were fed *ad libitum* on feeding days; therefore, starvation is unlikely (and all survived well following this experiment). It is possible that during this prolonged breeding season, females in good body condition had the capacity to invest their energy reserves into oogenesis to produce a second clutch, and thus leading to a more profound loss of mass. Male frogs also lost mass during the experiment, with over 20% loss of mass observed. This loss of mass in males could be due to the production of calling displays, which are energetically costly. Frogs in the breeding season will often forgo food and self-maintenance in order to maximize their breeding success (*Robertson, 1986*; *Ryser, 1989*), with some males losing over 30% of their body mass over the breeding season (*Robertson, 1986*).

## CONCLUSION

Captive breeding of understudied species can be challenging but is critically important to help protect declining or vulnerable species. In this study, we tested two methods of captive breeding on an endangered amphibian species that had never been bred in captivity: exogeneous hormone induction and the use of outdoor mesocosm enclosures. We found that outdoor breeding mesocosms with brumation were successful in producing viable egg clutches in *Litoria verreauxii alpina*, while exogenous hormones at the doses trialed here were not successful in producing any egg masses, viable or not. If outdoor conditions can loosely mimic the temperatures of the alpine region during *L. v. alpina*'s natural breeding season, then it could be an effective captive breeding strategy. Egg mass viability was fairly low, with 35% of clutches laid being unviable. Therefore, more work needs to be done to optimize breeding success.

## ACKNOWLEDGEMENTS

We would like to thank Thien Pham, Tara Jadwani-Bungar, Addison Zhou, Jess Wong, Alexander Wendt and the many research volunteers who assisted with animal husbandry over the two years of this project. We thank Marcus Hough and Anthony Waddle for assistance collecting and rearing the animals, and Lee Berger, Lee Skerratt, Simon Clulow, and Phillip Byrne for their advice.

### Funding

This work was supported by the Melbourne Veterinary School establishment grant, the Australian Research Centre Discovery Early Career Research Award (DE180101395) and Inlaks Shivdasani Foundation (Inlanks Ravi Sankaran Conservation Fellowship 2019). The funders had no role in study design, data collection and analysis, decision to publish, or preparation of the manuscript.

### Grant Disclosures

The following grant information was disclosed by the authors:
The Melbourne Veterinary School establishment grant.
The Australian Research Centre Discovery Early Career Research Award: DE180101395.
Inlaks Shivdasani Foundation (Inlanks Ravi Sankaran Conservation Fellowship 2019).

### Competing Interests

Laura A Brannelly is an Academic Editor for PeerJ.

### Author Contributions

- Laura A. Brannelly conceived and designed the experiments, performed the experiments, analyzed the data, prepared figures and/or tables, authored or reviewed drafts of the article, and approved the final draft.
- Preeti Sharma conceived and designed the experiments, performed the experiments, authored or reviewed drafts of the article, and approved the final draft.
- Danielle K. Wallace conceived and designed the experiments, performed the experiments, authored or reviewed drafts of the article, and approved the final draft.

### Animal Ethics

The following information was supplied relating to ethical approvals (i.e., approving body and any reference numbers):
    The University of Melbourne animal ethics

### Data Availability

    The raw data are available in the Supplementary File.

### Supplemental Information

Supplemental information for this article can be found online at http://dx.doi.org/10.7717/peerj.15179#supplemental-information.

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
