# Peer review of "Captive breeding in the endangered alpine tree frog, Litoria verreauxii alpina"

_PeerJ, doi:10.7717/peerj.15179_

## Round 0.1 · original submission · Major Revisions

· Academic Editor

Major Revisions

Dear Authors

Thank you for your submission. The manuscript has been reviewed and needs a major revision before further consideration. The reviewers have commented on every section needing improvement, particularly reviewer 2. I consider these comments are mandatory to put in the revised version. Some specific comments are below.

The misreporting of some of the trial period data for egg mass laying and the fact that some of the figures say one thing and the manuscript paragraphs say another does not appear malicious or of ill-intent by the authors, however, it does seem careless. Particularly that some figures and figure legends don't accurately match what is reported in the body of the manuscript. Many of the Arrive guidelines checklist sections/lines need additional details, particularly insufficient details on housing and husbandry which is key to the replication of their outdoor vs. indoor mesocosm trials of this study.

·

Basic reporting

This study provides valuable information on improving the captive breeding techniques for this declining species. This manuscript could be improved by providing additional background (lines 64-79) on why hormones are being used to induce breeding if this species has never been bred in captivity before. It seems that attempting to breed in captivity should begin with mimicking natural conditions as closely as possible, i.e., not artificially injecting hormones. Hormone treatments should only be attempted after unsuccessful breeding attempts. Also, the way I read this study implies that hormones were not used in the outdoor breeding experiment. It would help readers to confirm this and some explanation of why. Was it just because the original hormone treatments were unsuccessful at producing eggs?

Experimental design

This study provides timely information on refining breeding techniques for a declining species. As stated in the manuscript (lines 38-42) most species require specialized techniques to successfully breeding in captivity and this study addresses the alpine treefrog specifically, improving the knowledge base to captively breed this species. I think the methods are written well to describe the techniques used in this study.

Validity of the findings

This study presents novel data for a declining species that has not been bred in captivity before. This information is critically import for future conservation of this species and can be added to a growing literature base on captive breeding techniques for amphibians. The conclusions presented in this manuscript are clear, concise, and relevant to conservation of the alpine treefrog.

Additional comments

This study provides valuable information on captive breeding for the alpine treefrog. My experience working in with captive breeding programs for declining amphibian species in southern California has shown that each species has unique requirements for breeding and any additional information to improve breeding success can have huge positive impacts on their conservation.

Reviewer 2 ·

Basic reporting

Clear and unambiguous, professional English used throughout.
-No comment

Literature references, sufficient field background/context provided.
-There are areas where additional field background and context is needed; some areas where more citations are needed to support statements/facts provided by authors [see detailed annotated PDF for line edit recommendations]

Professional article structure, figures, tables. Raw data shared.
-This is the most critical area of basic reporting issues. While I thank the authors for sharing their raw data, some data on masses of individual frogs reported appears to be missing in the supplemental data from across the different trial periods. The supplemental data files need more descriptive metadata identifiers to be useful to and accurately interpreted by future readers. Figure legends 2 and 3 are reporting results differently from what was written in the methods and results paragraphs; Figure 3 indicates that the median mass of males and females were both lower after brumation than before it started and the individual points are also lower along the y-axis; however Line 227 contradicts Figure 3, stating that “Neither males nor females lost weight during their brumation period.”

Self-contained with relevant results to hypotheses.
-Hypotheses for the hormone and mesocosm experiments within can be more clearly stated in the introduction, particularly given the literature they cite of previous L. v. alpina hormone studies and what they might expect given prior mesocosm/captive breeding studies for Litoria sp.

Experimental design

Original primary research within Aims and Scope of the journal.
-No comment

Research question well defined, relevant & meaningful. It is stated how research fills an identified knowledge gap.
- No comment

Rigorous investigation performed to a high technical & ethical standard.
-See annotated PDF for detailed line-item comments on methodology and results in particular; some of the study data appears to be missing from the supplemental file or some portions of methodology accidentally misreported in the manuscript

Methods described with sufficient detail & information to replicate.
-This is the most important issue: particularly the design of the breeding enclosures and outdoor mesocosms; some of the experimental control groups; clear details of how many total individuals are in which experimental groups at which time points. A table may be a helpful tool for a reader to refer back to for all of the experimental groups/trials, indicating how many individuals of what sex were in each trial.
-There is information lacking as to the number of tadpoles originally collected in the wild, when they were collected, and a general collection site location (precise data understood to not be allowed since they are a vulnerable species).
-There was variation in the sex ratios for the breeding mesocosms trials, and they were changed around at Day 100 when male and female individuals were redistributed to promote spawning which was not clearly detailed.
-Methods for conducting estimation of total egg mass size/percentage viable should be detailed
-Methods for randomization of individuals to be put into which mesocosm groups and during the mesocosm redistribution should be detailed

Validity of the findings

Impact and novelty not assessed. Meaningful replication encouraged where rationale & benefit to literature is clearly stated.
-Impact and novelty somewhat assessed Lines 235-236: “ Litoria species are notoriously difficult to induce spawning in a captive setting (Clulow et al. 2018); therefore, this result was not surprising.”
-Authors encourage meaningful replication of mesocosm experiment with individuals that had not been bred/used in a prior breeding experiment; authors don’t encourage or describe how future experimentation might be improved to avoid the issues with communal mesocosms with linking individuals/breeding pairs to their subsequent clutches or infertile egg masses
-Authors do suggest additional hormone induction trials to determine the right dosage to invoke both breeding behavior and spermiation in male L. v. alpina

All underlying data have been provided; they are robust, statistically sound, & controlled.
-See annotated PDF for detailed line-item comments on the data provided regarding methodology and results; some reported data regarding masses appear to be missing from the supplemental file and some of the outdoor mesocosm egg data are misreported in the manuscript (e.g. total number of days for the outdoor mesocosm breeding trial, when the last)

Conclusions are well stated, linked to original research question & limited to supporting results.
-Yes, however, authors could give more specificity to their mesocosm success and hormone non-success more explicitly (see Lines 326-327)
-Authors support the result of success with outdoor breeding mesocosms; however they describe in Line 328 that if “outdoor conditions can loosely mimic the alpine region during L. v. alpina’s breeding season, then it could be an effective captive breeding strategy.” But the authors don’t describe in the introduction or discussion what the temperature and rainfall conditions are during L. v. alpina’s breeding season in situ in the Australian Alps so it will likely be difficult for future readers to make a useful comparison.

Additional comments

The need for information on this non-model species is critical and is currently lacking for captive breeding efforts and long-term conservation needs, and I applaud the authors for their novel captive breeding mesocosm success with this species. There are some primary weaknesses, including a lack of in situ environmental context to appropriately compare the mesocosm's effectiveness of mimicking their natural habitat; edits and additional review needed for accuracy in the way the study results are being reported; a need for more detailed methodology for experimental replication (particularly of mesocosm design)- all which should be improved upon before Acceptance.

Annotated reviews are not available for download in order to protect the identity of reviewers who chose to remain anonymous.

---

## Round 0.2 · Major Revisions

· Academic Editor

Major Revisions

Dear Authors
Thank you for your submission of revised version. It has been improved than the previous version but results are still need substantial revision/addition. The current results and analysis are not sufficient to consider as full length article. Some comments are below

Explicit details about the amphibian colony that the authors have denied providing in their rebuttal letter prior to the experimental start, like the number of tadpoles collected from the wild, and collection site details may seem- to the authors- to be irrelevant to their paper.
"These details are irrelevant to the study at hand and will be arduous for a reader to get through" Authors must provide the data to mention the journal that they should not publish the data.

However, these are the precise types of details when it comes to captive breeding efforts (and eventually reintroduction efforts) that tend to be dismissed initially as insignificant but are crucial for long-term success when it comes to replication [either for this species or another endangered species of frog that needs to be captive bred]. Is this species at risk of exploitation due to in-situ or ex-situ value or persecution? While I recognize that well-informed control of location data for highly sensitive taxa is necessary to avoid risks, such as poaching or habitat disturbance by recreational visitors, I would argue that ignoring the benefits of sharing basic demographic data for the individuals of this species used in this study could unnecessarily obstruct future conservation efforts for species and locations with lower risks of exploitation.

Add table or graphs more to make the results descriptive. Authors can do more analysis that are applicable to the collected data. Some comments of the reviewer are also mentioned.

Reviewer 2 ·

Basic reporting

No comment.

Experimental design

No comment.

Validity of the findings

Given the authors'comment in their rebuttal letter regarding their weights pre- and post- brumation and how it is visualized in their figure, would suggest Line 254 edited to "Neither males nor females lost a significant amount of weight during their brumation period." Thank you to the authors for addressing this in their letter, as well as other comments made for this reviewing category.

Additional comments

Environmental hydroperiods and rainfall are significant components of successful anuran breeding in the wild. While this study was focused on breeding ex-situ in a captive setting and exact replication of natural environments is very clearly not possible nor realistic to expect, to not record it in their natural habitat nor consider rainfall patterns seems like something that could be improved upon for future outdoor mesocosm studies that are meant to simulate/mimic their natural environment. I would strongly suggest that for future studies and mesocosm trials that the authors consider this factor and how it may impact captive breeding attempts for this species.

This manuscript is indeed greatly improved and I support its publication and acknowledge the authors' extensive revision efforts for their manuscript and submitted datasheet.

---

## Round 0.3 · accepted · Accept

· Academic Editor

Accept

Dear Authors

It is pleasure to inform you that the manuscript has been accepted based on the reviewers' comments and this manuscript is ready for publication.